# Spatial and temporal invasion dynamics of the 2014–2017 Zika and chikungunya epidemics in Colombia

Kelly Charniga[1]*, Zulma M. Cucunubá[1], Marcela Mercado[2], Franklyn Prieto[2], Martha Ospina[2], Pierre Nouvellet[3], Christl A. Donnelly[1,4]

1 Medical Research Council Centre for Global Infectious Disease Analysis, Department of Infectious Disease Epidemiology, Imperial College London, London, United Kingdom, 2 Instituto Nacional de Salud, Bogotá, Colombia, 3 School of Life Sciences, University of Sussex, Brighton, United Kingdom, 4 Department of Statistics, University of Oxford, Oxford, United Kingdom

* kelly.charniga@gmail.com

## Abstract

Zika virus (ZIKV) and chikungunya virus (CHIKV) were recently introduced into the Americas resulting in significant disease burdens. Understanding their spatial and temporal dynamics at the subnational level is key to informing surveillance and preparedness for future epidemics. We analyzed anonymized line list data on approximately 105,000 Zika virus disease and 412,000 chikungunya fever suspected and laboratory-confirmed cases during the 2014–2017 epidemics. We first determined the week of invasion in each city. Out of 1,122, 288 cities met criteria for epidemic invasion by ZIKV and 338 cities by CHIKV. We analyzed risk factors for invasion using linear and logistic regression models. We also estimated that the geographic origin of both epidemics was located in Barranquilla, north Colombia. We assessed the spatial and temporal invasion dynamics of both viruses to analyze transmission between cities using a suite of (i) gravity models, (ii) Stouffer's rank models, and (iii) radiation models with two types of distance metrics, geographic distance and travel time between cities. Invasion risk was best captured by a gravity model when accounting for geographic distance and intermediate levels of density dependence; Stouffer's rank model with geographic distance performed similarly well. Although a few long-distance invasion events occurred at the beginning of the epidemics, an estimated distance power of 1.7 (95% CrI: 1.5–2.0) from the gravity models suggests that spatial spread was primarily driven by short-distance transmission. Similarities between the epidemics were highlighted by jointly fitted models, which were preferred over individual models when the transmission intensity was allowed to vary across arboviruses. However, ZIKV spread considerably faster than CHIKV.

## Author summary

Understanding the spread of infectious diseases across space and time is critical for preparedness, designing interventions, and elucidating mechanisms underlying transmission.

**Data Availability Statement:** Data and code for reproducing the best-fitting gravity model results and the figures in the main text as well as the weekly time series at the city level are available on

GitHub (http://github.com/kcharniga/zika_chik_
invasion).

**Funding:** KC, ZMC, PN, and CAD acknowledge
funding from the MRC Centre for Global Infectious
Disease Analysis (reference MR/R015600/1),
jointly funded by the UK Medical Research Council
(MRC, https://mrc.ukri.org) and the UK Foreign,
Commonwealth & Development Office (FCDO,
https://bit.ly/3gw2uER), under the MRC/FCDO
Concordat agreement and is also part of the
EDCTP2 programme supported by the European
Union. KC is funded by Imperial College London's
President's PhD Scholarship (https://bit.ly/
3iMiXX1). ZMC is supported by a Fellowship
through the Rutherford Fund (MR/R024855/1,
https://bit.ly/3q7Bi2n). The funders had no role in
study design, data collection and analysis, decision
to publish, or preparation of the manuscript.

**Competing interests:** The authors have declared
that no competing interests exist.

We analyzed human case data from over 500,000 reported cases to investigate the spread of the recent Zika virus (ZIKV) and chikungunya virus (CHIKV) epidemics in Colombia. Both viruses were introduced into northern Colombia. We found that gravity models and Stouffer's rank models best described transmission and that transmission mainly occurred over short distances. Our results highlight similarities and key differences between the ZIKV and CHIKV epidemics in Colombia, which can be used to anticipate future epidemic waves and prioritize cities for active surveillance and targeted interventions.

## Introduction

The global burden of disease due to arboviral infections is substantial and continues to increase [1]. Chikungunya virus (CHIKV) is an alphavirus that is transmitted to people primarily by *Aedes* mosquitoes [2]. Symptoms of chikungunya fever, the disease caused by CHIKV, include fever, rash, and headache as well as intense joint pain, which can persist for weeks or months [3]. Cases of chikungunya fever were first reported in the Americas in December 2013 [3]. Within a year, over one million cases were reported in the region, including severe cases and deaths [4]. Zika virus (ZIKV) is a flavivirus that is also spread by *Aedes* mosquitoes. Symptoms of ZIKV disease resemble those of chikungunya fever but are typically milder [5]. In May 2015, Brazil became the first country in the Americas to detect cases of ZIKV disease. In October 2015, Brazil reported an association between ZIKV infection during pregnancy and microcephaly, a birth defect characterized by head size that is smaller than expected based on age and sex. By February 2016, the World Health Organization declared the cluster of microcephaly and other neurological complications reported in Brazil a Public Health Emergency of International Concern [6]. From Brazil, ZIKV spread widely throughout Latin America and the Caribbean. There are currently no approved drugs to treat or prevent ZIKV disease or chikungunya fever, although several vaccine candidates are under investigation [7,8].

Previously, the spatial and temporal spread of ZIKV [9–11] and CHIKV [12–14] in the Americas has been studied separately. However, the viruses share common vectors and were both introduced into apparently immunologically naïve populations. An integrated study of these diseases in the same country may help elucidate similarities and differences between the two. Here, a suite of spatial interaction models, including variations of the gravity model, Stouffer's rank model, and radiation model, were fitted to analyze transmission between cities in Colombia, one of the countries most affected by the ZIKV and CHIKV epidemics in the Americas [15,16]. There is a high risk of major infectious diseases in Colombia, including bacterial diarrhea from food or water as well as vector-borne diseases [17]. Each year, the country is faced with dengue virus epidemics caused by one or more of the four known viral serotypes. Like ZIKV, dengue virus is a flavivirus spread by *Aedes* mosquitoes; it is the causative agent of dengue fever. Malaria transmission is also recorded annually in Colombia, with epidemic cycles of two to seven years [18]. In this study, the invasion dynamics of CHIKV and ZIKV were examined as well as the extent to which inter-city transmission depended on distance, population sizes of invaded and susceptible cities, and the infectivity of each virus.

## Results

### Temporal and spatial patterns in invasion weeks

Out of 1,122 cities in Colombia, week of invasion was determined for 338 cities for CHIKV and 288 cities for ZIKV. Invasion weeks ranged from the week ending May 31, 2014 to that

**Table 1. Epidemiological characteristics of CHIKV and ZIKV epidemics in Colombia.**

| | Cities (#) | Time for spread to 50%* (weeks) | Time for spread to 100%** (weeks) | Calendar time for 90% of spread*** | Long-distance transmission events**** (#) |
|---|---|---|---|---|---|
| **CHIKV** | 338 | 31 | 68 | Sept. 2014-May 2015 (35 weeks) | 4 |
| **ZIKV** | 288 | 16 | 33 | Sept. 2015- Jan. 2016 (21 weeks) | 3 |

*Time for 50% of cities to be invaded.

**Time from the first city to be invaded to the last city to be invaded.

***Calendar time for 90% of cities to be invaded (5th percentile to 95th percentile).

****More than 344.4 km for CHIKV and more than 321.21 km for ZIKV. See Methods and S1 Text.

ending September 19, 2015 for CHIKV and from the week ending August 8, 2015 to that ending March 26, 2016 for ZIKV. The time for the diseases to invade 50% of cities ever affected was shorter for ZIKV compared to CHIKV (Table 1), and while invasion weeks for ZIKV tended to cluster within five months (from September 2015 to January 2016), 90% of invasion weeks for CHIKV clustered within nine months (between September 2014 and May 2015). For cities that experienced epidemics of both CHIKV and ZIKV (n = 205), invasion weeks were significantly positively correlated (Pearson's correlation coefficient 0.45, $p < 0.0001$). Both epidemics were first recorded in northern Colombia and spread from there. Early foci of disease were also present in the central parts of the country (Fig 1).

## Geographic origin of epidemics

The first city to report cases of chikungunya fever in Colombia was Planeta Rica, Córdoba. Cases of ZIKV disease were first reported in the country simultaneously by five cities: (i) Cali, Valle del Cauca, (ii) San Andrés, San Andrés and Providencia, (iii) Cúcuta, Norte de Santander, (iv) El Zulia, Norte de Santander, and (v) Puerto Santander, Norte de Santander. Assuming a linear relationship between invasion week and geographic distance from the source of the epidemic, the estimated origin of both epidemics was Barranquilla, Colombia's fourth most populated city located on the Caribbean coast (Fig 2). According to the line list data, Barranquilla was among the first five cities to report cases of chikungunya fever, first reporting cases in week 12 (invaded in week 11). The city was also among the first 18 cities to report cases of ZIKV disease, first reporting cases in week 5 (invaded in week 4).

Fig 3 shows the spread of reported cases during the epidemics. Epidemiological curves by department can be found in S1 Text. Most departments had some overlap in reported chikungunya fever cases and ZIKV disease cases. Although none of the departments had a peak in the incidence of both diseases at the same time, only eight weeks separated the peaks of the CHIKV and ZIKV epidemics in Putumayo. S1 and S2 Movies show the monthly incidence per 100,000 population by first administrative unit (department).

## Long-distance transmission events

Four long-distance transmission events were identified for CHIKV and three were identified for ZIKV. For CHIKV, the affected cities were (i) Girardot, Cundinamarca, (ii) La Primavera, Vichada, (iii) Mocoa, Putumayo, and (iv) Puerto Asís, Putumayo. For ZIKV, the affected cities were (i) Barranquilla, Atlántico, (ii) Tauramena, Casanare, and (iii) Cartagena, Bolívar. Three of these seven cities are department capitals. All of these events occurred early in the epidemics, within the first 15% of cities invaded (Methods and S1 Text). Long-distance transmission events for CHIKV occurred at distances of 366, 402, 431, and 475 km compared to a mean distance of 25.7 km. Long-distance transmission events for ZIKV occurred at distances of 322,

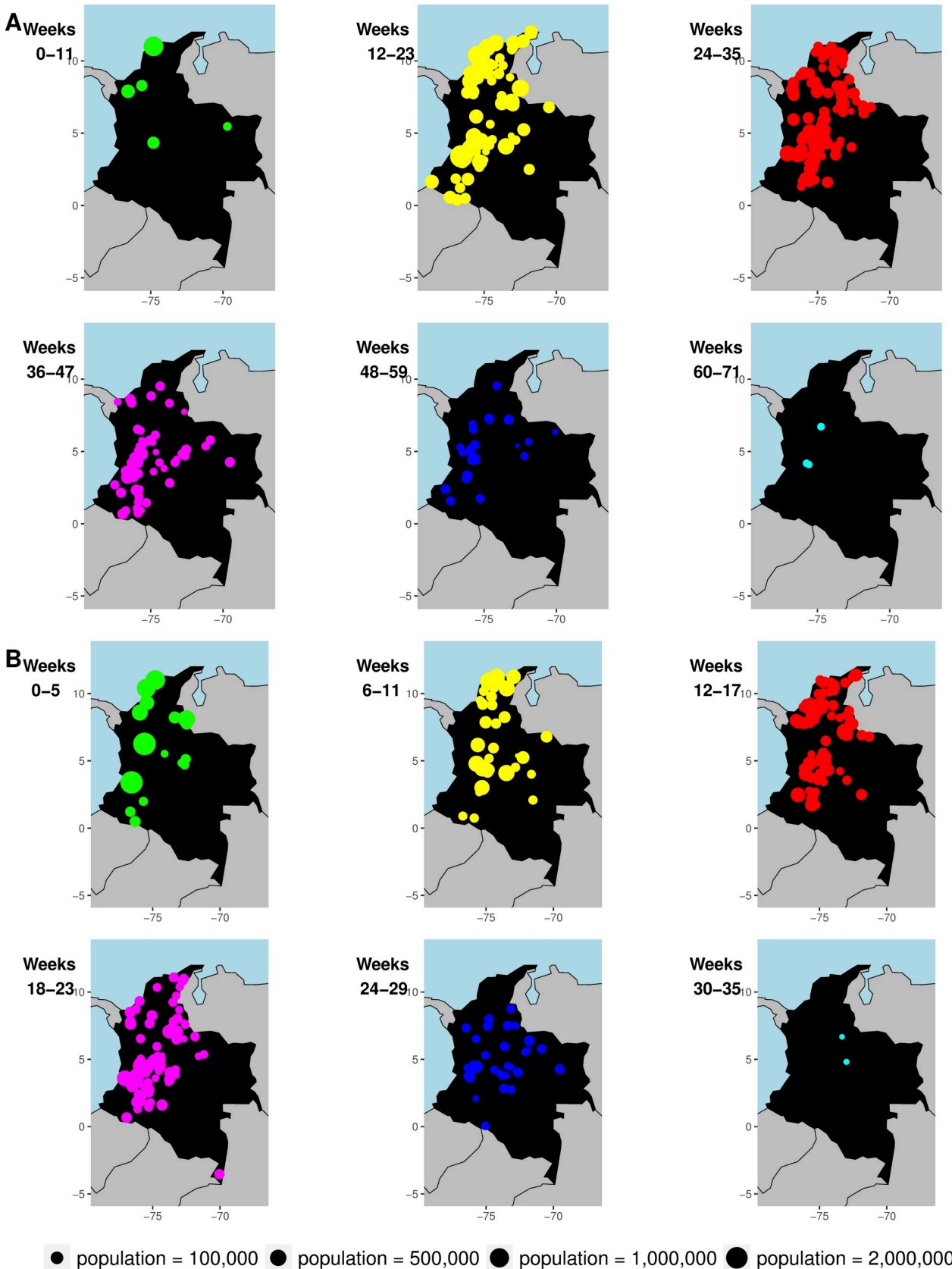

**Fig 1. Geographic patterns of invasion weeks in studied cities in Colombia based on first reported cases.** Invasion weeks are shown by 12-week period for (A) CHIKV and 6-week period for (B) ZIKV. Each circle represents a city, and the size of the circle is proportional to population size. Each panel shows only cities newly invaded during the time period indicated in the upper left-hand-corner. The island of San Andrés is not shown but was invaded by CHIKV in week 21 and by ZIKV in week 0. Made with Natural Earth. Free vector and raster map data @ naturalearthdata.com.

336, and 346 km compared to a mean distance of 25.6 km. Potential sources of the affected cities can be found in S1 Text.

It is important to note that the methods for estimating the epidemic origin and long-distance transmission events are independent of one another. In the case of ZIKV, for example, Barranquilla is estimated as both the epidemic origin and a long-distance transmission event.

## Spatial interaction models

**Models fitted independently to each virus.** Models were initially fitted to cities that had available data on both distance metrics (337 and 287 cities for CHIKV and ZIKV, respectively). The best-fitting CHIKV model was Stouffer's rank model with geographic distance (S1 Text). The next best-fitting models were Stouffer's rank model and a version of the gravity model that incorporates spatial interaction (also known as Fotheringham's competing destinations model [19]), both fitted to travel time between cities. The change in Deviance Information Criterion (DIC) among the first three models was not meaningful ($\leq 4$). The fourth best-fitting model was a gravity model (competing destinations version) incorporating geographic distance, with a change in DIC of 6.5 compared to the best-fitting model. Although some models

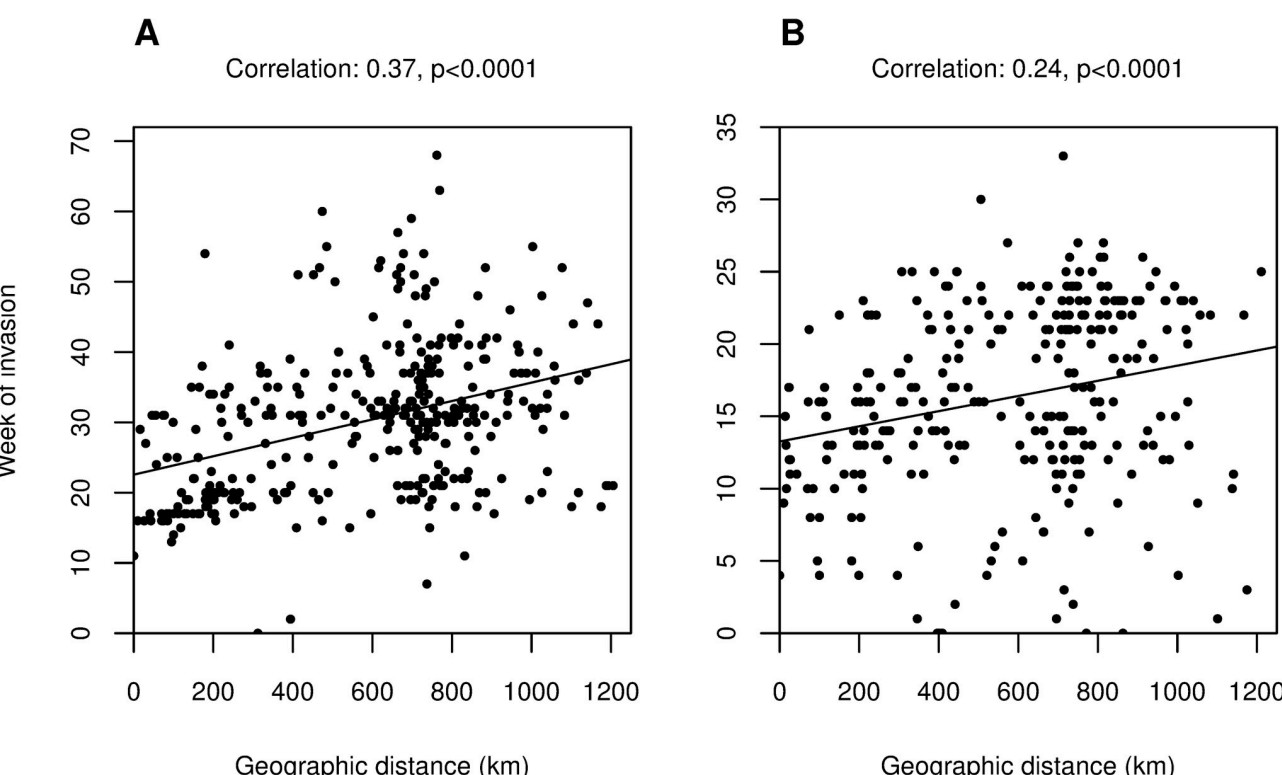

**Fig 2. Correlations between city invasion weeks and geographic distance from first invaded cities for CHIKV and ZIKV.** Week of invasion for each invaded city is shown on the y-axis for both plots. These weeks are plotted against (A) the geographic distance from the most likely origin of CHIKV in Colombia, Barranquilla and (B) the geographic distance from the most likely origin of ZIKV in Colombia, also Barranquilla. Pearson's correlation coefficients and significance are shown above each plot.

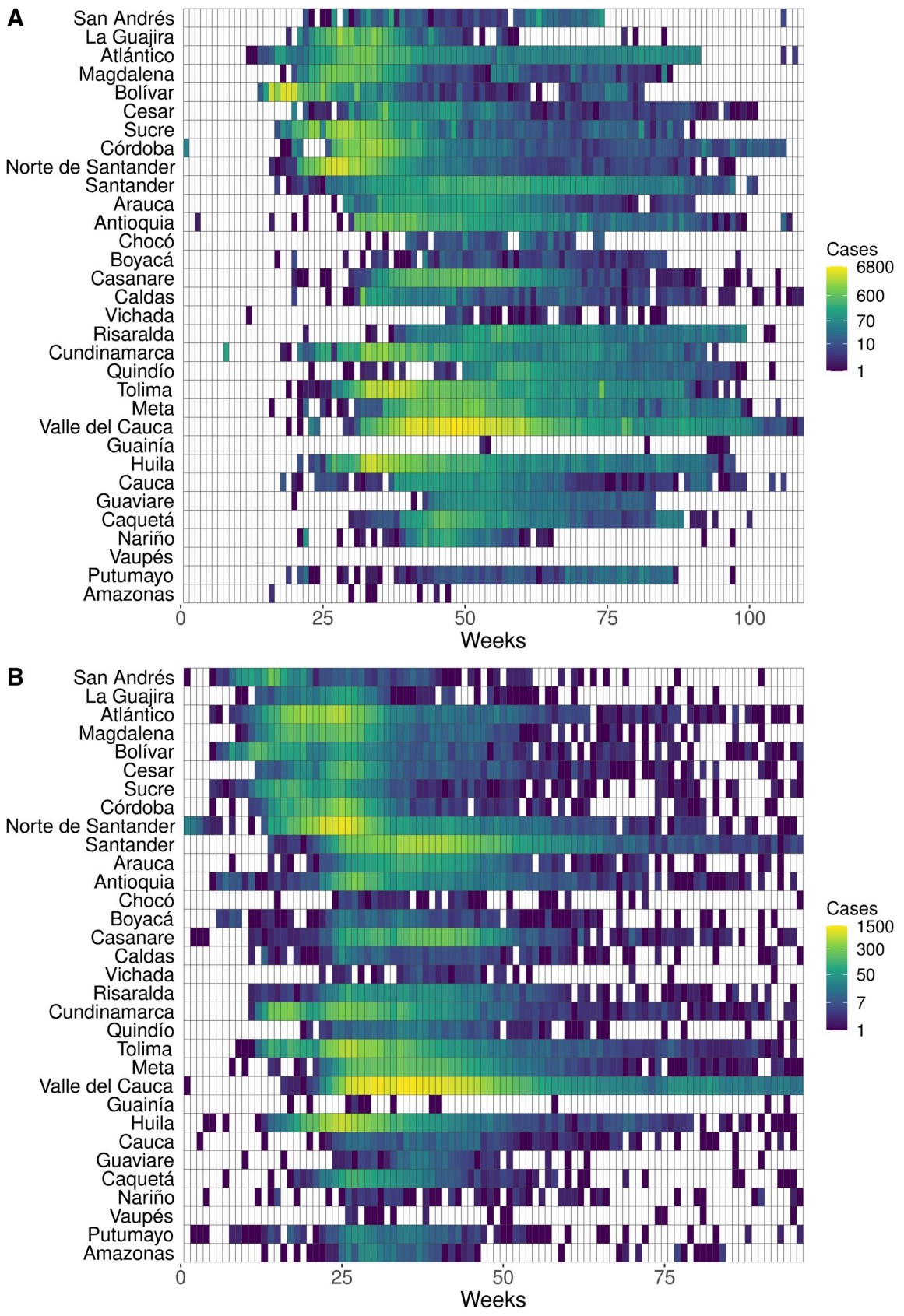

**Fig 3. Heatmaps showing the spatial and temporal spread of CHIKV and ZIKV in Colombia.** Population-weighted centroids were used to rank departments in order from North to South. Colors across rows represent the number of cases of (A) chikungunya fever and (B) ZIKV disease for each department. Weeks are plotted on the x-axis starting from the first week cases were reported to the last week cases were reported. Dates for (A) range from the week ending June 7, 2014 to that ending July 9, 2016, and dates for (B) range from the week ending August 15, 2015 to that ending June 17, 2017. White rectangles are weeks with zero reported cases.

fitted to travel time between cities had lower DIC values than the same model type fitted to geographic distance, the difference was only meaningful for the radiation variant model. The radiation and radiation variant models performed the least well.

In contrast to CHIKV, the best-fitting model for ZIKV was the competing destinations version of the gravity model with geographic distance (S1 Text). This model was followed by Stouffer's rank model and Stouffer's rank variant model, both incorporating geographic distance. Within model types, ZIKV models fitted to geographic distance were preferred over those fitted to travel time between cities. As with CHIKV, the models that performed least well were the radiation and radiation variant models. Other versions of the gravity models can be found in S1 Text.

For both epidemics, the best-fitting gravity model (based on the lowest DIC) included the following parameters: a distance power, power for invaded city population size, density dependence, infectivity, and transmission intensity. In both instances, the population size of the susceptible city appeared uncorrelated with the invasion dynamics. The estimated distance power, $\gamma$, for each model was 1.68 (95% credible interval [CrI]: 1.44–1.90) for CHIKV and 1.74 (95% CrI: 1.51–1.96) for ZIKV. Thus, we cannot exclude the possibility that the relationship with distance was the same for both viruses. Both models also estimated intermediate levels of density dependence.

In contrast, invasion risk was associated with the population size of the susceptible city in both Stouffer's rank models. Estimates of the infectivity parameter were similar to those obtained from the gravity models. Although the estimates of transmission intensity were lower in the Stouffer's rank models, ZIKV still had a higher estimate compared to CHIKV. Estimates of the effect of invaded city population size were stronger; however, because this parameter additionally captures spatial interaction, the interpretation is different compared to the gravity models.

**Models fitted jointly to arboviruses.**   The four model variants selected for the joint analysis included the gravity model and Stouffer's rank model, each with either geographic distance or travel time between cities. The individual and joint models for the two best-fitting model variants are shown in Table 2 (gravity model with geographic distance and Stouffer's rank model with geographic distance); the difference in the individual models' sum of DIC was only 2.4. The third and fourth best-fitting individual and joint models are shown in S1 Text. We found that for each model variant considered, the joint model which assumed the same parameters across arboviruses had a higher DIC (i.e. worse fit) than the sum of the individual models' DIC. The fit of the joint models improved when the parameter for transmission intensity was allowed to vary across arboviruses. When both transmission intensity and infectivity parameters were allowed to vary across arboviruses, the DIC values of the joint models were only 1–2 units away from the individual models' summed DIC. Overall, the most parsimonious model with the lowest DIC was the joint gravity model with geographic distance and two parameters for transmission intensity (Table 2).

## Validation of gravity model fit and parameter fitting procedure

Model validation was performed for each individual virus' best-fitting gravity model with geographic distance. As geographic distance data were available for all cities, we used models fitted to 338 and 288 cities for CHIKV and ZIKV, respectively. The parameter values for these models can be found in S1 Text. For each city, we evaluated the predicted invasion week given the observed invasion weeks in other cities up to that time. The best-fitting models predicted the

**Table 2. Comparison of individual versus joint models of CHIKV and ZIKV spread in Colombia.**

| | Model type[*] | Distance type[**] | DIC | Sum of DIC | γ (distance power) | μ (susceptible population)[***] | ν (invaded population) | ε (spatial interaction) | φ_a (infectivity) | φ_b (infectivity) | β_a (intensity) | β_b (intensity) |
|---|---|---|---|---|---|---|---|---|---|---|---|---|
| CHIKV | G | GD | 2329.4 | 4044.9 | 1.68 (1.44–1.90) | 0 | 0.65 (0.53–0.76) | 0.83 (0.68–0.99) | 0.35 (0.25–0.46) | | 0.24 (0.14–0.39) | |
| ZIKV | G | GD | 1715.5 | | 1.74 (1.50–1.97) | 0 | 0.55 (0.41–0.69) | 0.67 (0.50–0.83) | 0.27 (0.13–0.40) | | 1.11 (0.68–1.81) | |
| Joint | G | GD | | 4129.0 | 1.69 (1.54–1.84) | 0 | 0.55 (0.47–0.64) | 0.81 (0.71–0.92) | 0.12 (0.07–0.16) | | 0.52 (0.39–0.70) | |
| Joint | G | GD | | 4042.6 | 1.72 (1.56–1.89) | 0 | 0.61 (0.52–0.70) | 0.76 (0.65–0.87) | 0.32 (0.24–0.40) | | 0.31 (0.21–0.46) | 0.90 (0.65–1.28) |
| Joint | G | GD | | 4043.3 | 1.71 (1.56–1.87) | 0 | 0.60 (0.51–0.69) | 0.76 (0.65–0.86) | 0.35 (0.25–0.45) | 0.27 (0.14–0.40) | 0.28 (0.18–0.44) | 0.91 (0.66–1.24) |
| CHIKV | S | GD | 2322.9 | 4047.3 | | 0.48 (0.37–0.58) | 1.18 (1.01–1.36) | | 0.32 (0.24–0.42) | | 0.009 (0.005–0.015) | |
| ZIKV | S | GD | 1724.4 | | | 0.43 (0.31–0.55) | 1.37 (1.12–1.63) | | 0.53 (0.44–0.63) | | 0.021 (0.013–0.032) | |
| Joint | S | GD | | 4143.5 | | 0.43 (0.36–0.51) | 1.31 (1.15–1.46) | | 0.21 (0.17–0.25) | | 0.022 (0.016–0.028) | |
| Joint | S | GD | | 4055.5 | | 0.44 (0.36–0.51) | 1.18 (1.03–1.33) | | 0.41 (0.34–0.49) | | 0.006 (0.004–0.010) | 0.020 (0.014–0.028) |
| Joint | S | GD | | 4046.3 | | 0.45 (0.38–0.53) | 1.25 (1.09–1.40) | | 0.31 (0.23–0.40) | 0.54 (0.44–0.63) | 0.011 (0.006–0.017) | 0.017 (0.012–0.024) |

Posterior median and 95% credible interval presented for each parameter. Models were fitted to 337 cities for CHIKV and 287 cities for ZIKV. For the joint models that estimate different parameters across arboviruses, parameters with subscript a refer to CHIKV, while parameters with subscript b refer to ZIKV.

[*]G: gravity (competing destinations), S: Stouffer's rank

[**]GD: geographic distance

[***]When μ is set to 0, this means that cities with large populations have the same risk of being invaded as cities with small populations.

distribution of the local start of epidemics well (Fig 4). Excluding cities that were invaded in week 0, for CHIKV 304 out of 337 cities (90% of cities, 95% CI: 87–93%) lie within the 95% interval of their expected distribution, and for ZIKV 268 out of 283 cities (95% of cities, 95% CI: 91–97%) lie within the 95% interval of their expected distribution. Cities that fell outside of these intervals tended to be invaded at the beginning or the end of the epidemics (S1 Text). The best-fitting ZIKV model captured the shape of the observed invasion week distribution well. In contrast, the best-fitting CHIKV model did not capture the shape well at the end of the epidemic. Cities invaded late in the CHIKV epidemic (week 53 or later) had smaller population sizes and fewer cases compared to cities invaded earlier (up to week 52) (Wilcoxon rank-sum tests for population size: W = 885, p = 0.004 and for cumulative case numbers: W = 900, p = 0.005). However, these 11 late-invaded cities represent a small proportion of all invaded cities (3%).

Simulated epidemics from the best-fitting model for each virus were consistent with the observed epidemic in terms of the number of invaded cities over time (Fig 5). For CHIKV, we

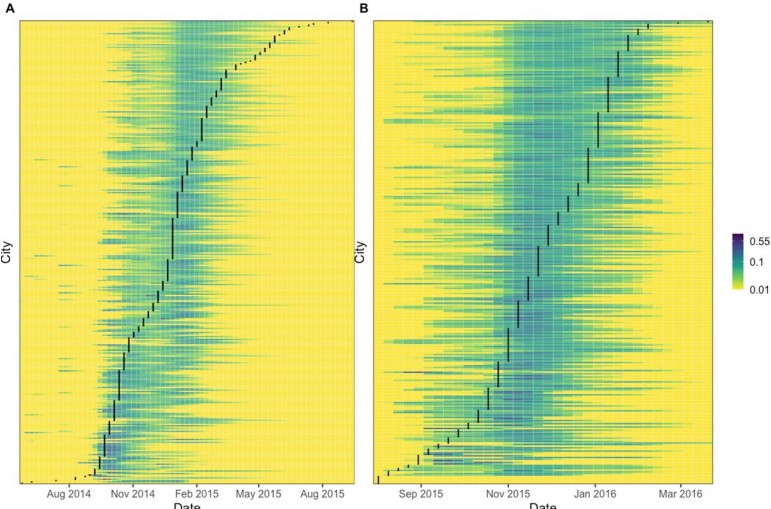

**Fig 4. Probability distribution of invasion weeks.** The panels show the estimated probability distributions of invasion week for each city (colored lines) for (A) CHIKV and (B) ZIKV based on the observed start of invasion in other cities up to that time. The calculations were performed using the median parameter estimates from the posterior distributions of the best-fitting models for CHIKV and ZIKV. The black lines show the observed invasion week based on the first reported cases in each city. Values plotted as 0.01 represent probabilities of 0.01 or less.

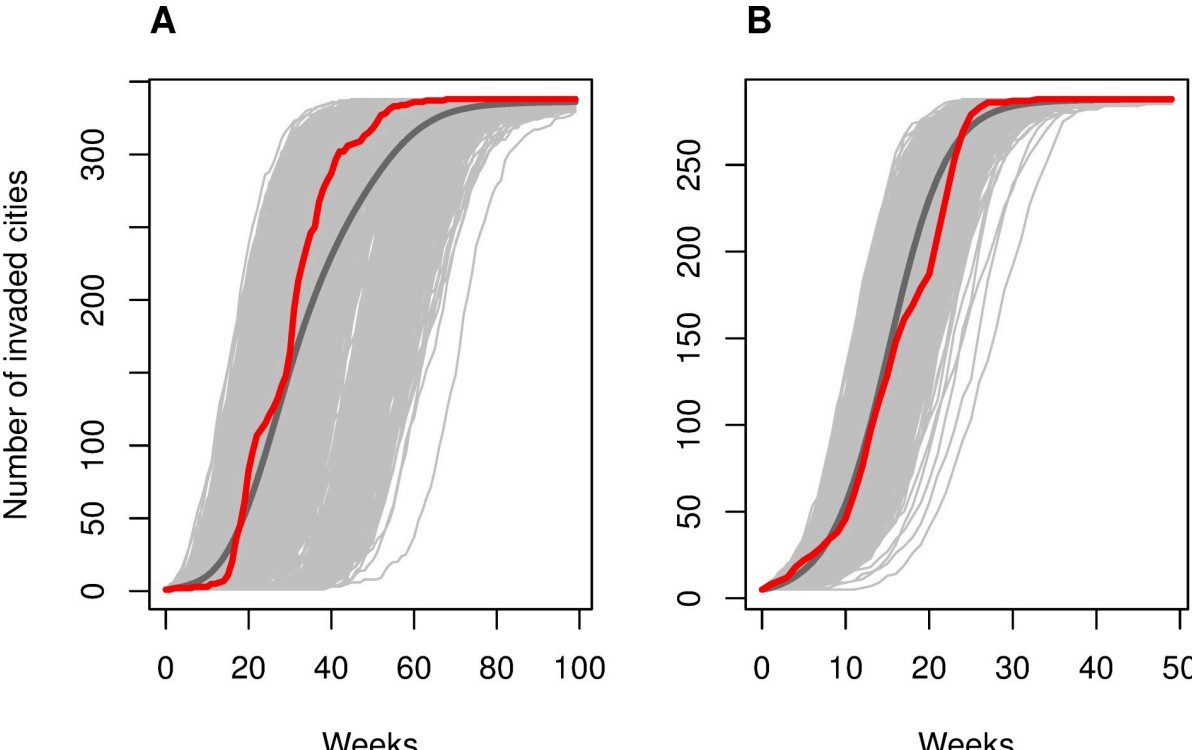

**Fig 5. Epidemic invasion simulations.** Simulated invasion week (as week of first reported cases) for (A) CHIKV and (B) ZIKV from the best-fitting models. Simulated epidemics are shown in light gray. The dark gray lines are the average across the 1,000 simulations. The red lines show the observed number of cities that first reported cases in each week.

observed half of the cities invaded by week 31 of the epidemic, while 1,000 simulations predicted half of the cities invaded by week 34.4 on average (min. 17, mode 27, max. 66). For ZIKV, we observed half of the cities invaded by week 16, while simulations predicted 15.7 on average (min. 11, mode 15, max. 23). Epidemic simulations showed that model fits were robust to the choice of threshold for invasion (S1 Text).

For each virus, we were able to recover the fitted parameter estimates from a model fitted to a single simulated dataset created by simulating the epidemic from the median parameter estimates. The results of the simulation study can be found in S1 Text.

Results of models fitted to all 1,122 cities in Colombia are presented in S1 Text. When all cities were included, as expected the transmission intensity estimate was much lower, and the epidemic simulations showed a very delayed and prolonged epidemic compared to the observed incidence of invaded cities.

We relaxed the single-introduction assumption for CHIKV to test the effect on parameter estimates following Eggo et al. [20]. By starting parameter estimation from week 12, we allowed five cities to seed the epidemic rather than one. The parameter estimate of the distance kernel was slightly higher, but the credible intervals largely overlapped, suggesting that this assumption does not greatly affect the model fit (S1 Text).

## Validation of Stouffer's rank model fit

We also performed epidemic simulations for the best-fitting Stouffer's rank models. Even though this model had a lower DIC than the best-fitting gravity model for CHIKV, the epidemic simulations were worse overall (S1 Text). In contrast, simulations for ZIKV were comparable across model types.

## Risk factors of invasion

For CHIKV, the following predictors of city invasion were significant at the 0.05 level in the univariate analysis: population size, elevation, mean temperature during the study period, mean temperature up to the epidemic peak, mean rainfall over the study period, mean rainfall up to the epidemic peak, percentage of households with overcrowding, percentage of households with inadequate exterior walls, and risk of dengue virus. Except for mean rainfall up to the epidemic peak, the other eight predictors were also significantly associated with invasion by ZIKV. In addition, the percentage of households with inadequate floors was almost significant (p = 0.06) (S1 Text).

Four variables were included in the best-fitting logistic regression model for CHIKV invasion: mean temperature during the study period, mean rainfall during the study period, dengue risk, and mean travel time. Both higher rainfall and longer travel time were protective for invasion. In contrast, temperature and dengue risk were associated with increased odds of invasion. The odds of invasion by CHIKV were 15.5 (95% CI: 7.39–34.84) times higher among cities in the third tertile of dengue risk compared to cities with no risk of dengue adjusting for other variables in the model. In the model where weather covariates were defined up to the epidemic peak rather than during the study period, the odds ratio for temperature was about the same (1.25), while the odds ratio for rainfall decreased slightly to 0.86. Four variables were also included in the best-fitting logistic regression model for ZIKV invasion: elevation, mean rainfall during the study period, and inadequate exterior walls were all protective for invasion, while dengue risk was associated with an increase in the odds of ZIKV invasion. The odds of invasion were 42.3 (95% CI: 16.0–135.4) times higher among cities in the third tertile of dengue risk compared to cities with no risk of dengue adjusting for other variables in the model (S1 Text). In the model where weather covariates were defined up to the epidemic peak, the

odds ratio for rainfall increased slightly to 0.84. There was no evidence of poor fit for either model according to the Hosmer and Lemeshow goodness-of-fit test (CHIKV: p = 0.56, ZIKV: p = 0.40).

Using linear regression models, we found that the time to invasion decreased by 3.4 (95% CI: 2.1–4.8) weeks for CHIKV and by 2.3 (95% CI: 1.3–3.4) weeks for ZIKV on average for each one-unit difference in dengue risk level (S1 Text).

## Discussion

Similarities and key differences in the space-time dynamics of the CHIKV and ZIKV epidemics in Colombia were identified using spatial interaction models. Spatial invasion of both epidemics likely began in the north (Caribbean region). From there, the Andes Mountains may have delayed epidemic spread southwards by serving as a natural barrier to human movement. The best-fitting models for each virus were different, and the ZIKV epidemic spread twice as fast as the CHIKV epidemic. Our gravity model parameter estimates for $\gamma$, $\nu$, $\varepsilon$, and $\phi$, characterizing the effects of distance, invaded city population size, density dependence, and infectivity, respectively, were consistent with those obtained in studies of seasonal and pandemic influenza spread [20,21]. Similarly, our parameter estimates for the effect of susceptible city population size, $\mu$, and the effect of population sizes of invaded and intervening cities, $\nu$, obtained from Stouffer's rank models were consistent with those obtained in a study of measles [22]. Cities with high historical dengue virus transmission had greater odds of being invaded compared to cities with no risk of dengue, and higher levels of dengue risk were associated with decreased time to invasion.

### Comparing alternative spatial interaction models

Across model types, geographic distance was the preferred distance metric to describe spread of ZIKV; in contrast, geographic distance described spread similarly to travel time between cities for CHIKV. Although Viboud et al. found that work commutes better described the spread of seasonal influenza in the US compared to geographic distance, Charu et al. found that models with geographic distance outperformed those using work commutes or air traffic [21,23]. Geographic distance was also a better predictor of CHIKV spread in the Caribbean region than air traffic [12].

Unlike the gravity model, Stouffer's rank model does not have a parameter for the effect of distance on disease spread. Rather, the $\nu$ parameter indirectly captures distance by accounting for the population sizes of cities located in between invaded and susceptible cities as well as the population size of the invaded city. Using Stouffer's rank models, we obtained similar estimates of $\mu$ for CHIKV and ZIKV (0.48 [95% CrI: 0.37–0.58] and 0.43 [95% CrI: 0.32–0.55], respectively). The credible intervals for $\nu$ also overlapped (CHIKV: 1.19 [95% CrI: 1.01–1.36], ZIKV: 1.37 [95% CrI: 1.15–1.65]). While our estimates for $\nu$ are similar to those reported by Bjørnstad et al. in their investigation of measles in England and Wales from 1944–1965, our estimates of $\mu$ were lower; they reported 1.44 for $\nu$ and 0.82 for $\mu$. They also found that Stouffer's rank model performed the best, followed by an extended version of the radiation model and the competing destinations model. They recommended that more than one class of models should be explored when attempting to predict the spatial spread of infectious diseases [22]. Kraemer et al. found that when used together, gravity and radiation models helped explain heterogeneity in the invasion process of Ebola virus in West Africa during the 2014–2016 epidemic [24].

Epidemic simulations of Stouffer's rank models were unable to reproduce the CHIKV epidemic, despite this model having a lower DIC than the best-fitting gravity model. Although

Stouffer's rank model was better able to capture the beginning of the epidemic, the gravity model performed better in the middle and at the end of the epidemic. A possible reason for this finding is that radiation-type models tend to capture commuting patterns, while gravity models are better suited toward longer-distance movements [24].

## Gravity models

The estimated power of the effect of distance on spread, $\gamma$, with geographic distance was about 1.7, indicating that transmission was dominated by short-distance interactions. Slightly higher estimates of about 2.0 were obtained for models using travel time between cities (S1 Text). We expected similar estimates of $\gamma$ for CHIKV and ZIKV because they were spread by the same vectors in the same geographic area. Using geolocated genotype and serotype data, Salje et al. found evidence that in Bangkok, Thailand, transmission of dengue virus is highly focal, with the majority of infection events occurring near the home [25].

A range of estimates of $\gamma$ have been reported in the literature. Gog et al. reported 2.6 (95% CI: 2.3–2.8) for 2009 pandemic influenza in the US, and Charu et al. reported a median of 2.2 (range 2.1–2.7 with standard deviations between 0.13 and 0.33) across seven influenza seasons in the US [21,26]. However, Eggo et al. reported lower values of 1.2 and 0.86 for 1918 pandemic influenza in England and Wales and in the US, respectively [20]. Differences could be attributed, at least in part, to data being aggregated at different spatial scales; fewer data points in an area will lead to lower estimates of the distance power because locations are farther apart.

For the best-fitting CHIKV and ZIKV models, we obtained similar estimates for the estimated power for the effect of invaded city population size $\nu$, indicating that cities with large populations are more likely to spread disease than cities with smaller populations. Gog et al. did not include this parameter, and in Eggo et al., it was not selected in their best-fitting England and Wales model.

For both CHIKV and ZIKV, we accepted the null hypotheses that the estimated powers ($\mu$) for the effects of susceptible city population size were 0. This means that cities with large populations have the same risk of being invaded as cities with small populations. However, $\mu$ did appear to contribute to the fit of the Stouffer's rank models, and therefore we cannot say definitively that susceptible city population size was unimportant in the spread of CHIKV and ZIKV. Low, but significant, estimates of $\mu$ were reported by Gog et al. (0.27, 95% CI: 0.11–0.44) and Eggo et al. (0.40, 95% CrI: 0.25–0.54) for seasonal and pandemic influenza, respectively.

Intermediate levels of density dependence best described transmission ($\epsilon$ for CHIKV 0.83, 95% CrI: 0.69–0.98 and ZIKV 0.68, 95% CrI: 0.50–0.84) (S1 Text). In other words, connectivity somewhat depended on the number and size of neighboring populations. For influenza in the US, Gog et al. and Charu et al. reported estimates of $\epsilon$ close to 1 [21,26]. Eggo et al. reported $\epsilon$ close to 1 for influenza in England and Wales but also found that a density-dependent model ($\epsilon = 0$) fit the data best for influenza in the US [20]. Similarly, Salje et al. found that dengue virus transmission in Bangkok, Thailand was consistent with density-dependent transmission ($\epsilon = 0$) [25]. Differences in estimates could be due to differences in both coverage of datasets and spatial scales considered.

We obtained low estimates for the infectivity parameter $\phi$ for both CHIKV (0.35, 95% CrI: 0.25–0.48) and ZIKV (0.27, 95% CrI: 0.13–0.40) models (S1 Text). This suggests that cities with more reported cases were more infectious than cities with fewer reported cases. One reason for low, though significant, estimates could be because reported case incidence poorly reflects the true incidence of infection in a city. For example, if reporting rates vary over time or by location, cases reported to the surveillance system may not be a good proxy for actual

infection incidence. Using mortality rate as a proxy for infectiousness, Eggo et al. reported a similar estimate of 0.24 (95% CrI: 0.03–0.47) for pandemic influenza in England and Wales [20].

Transmission intensity clearly differs between the two viruses. The estimated β for ZIKV is significantly higher than that for CHIKV, reflecting the faster spread of the ZIKV epidemic. Differences in transmission intensity could be related to the 2015–2016 El Niño weather phenomenon. Caminade et al. found that increased temperatures associated with El Niño created conditions across South America that were favorable for ZIKV transmission in 2015 [27].

## Joint models

Joint models of CHIKV and ZIKV were preferred over models fitted to each virus separately when parameters for transmission intensity and infectivity were allowed to vary across viruses. This finding suggests that some aspects of the spatiotemporal patterns of epidemic arboviruses in Colombia were the same.

## Conclusions and limitations

The results presented here depend on estimates of invasion week in each city. We defined invasion week as the week before cases were first reported in each city. At the beginning of an outbreak, one or even a small number of reported cases in a city may not be sufficient to sustain chains of transmission resulting in spread to other cities; however, the first reported cases could be the indication of previously undetected transmission. A genomic epidemiological study found evidence that ZIKV had been circulating undetected in Colombia for five to eight months before the first cases were confirmed in September 2015 [28]. Moreover, because we modeled the city of likely infection rather than city of notification or residence (S1 Text), it is possible that cities with better surveillance or healthcare infrastructure could have been the first to report cases in travelers returning from cities with no previous evidence of transmission.

Our results are robust to uncertainty in invasion weeks. We fitted the models using an alternative definition for invasion week (S1 Text). Although model fits for CHIKV were slightly worse, ZIKV model fits were comparable. Reassuringly, parameter estimates were similar (S1 Text). Our results are also robust to the choice of threshold for the number of reported cases. For each virus, gravity model simulations were similar for thresholds of 10, 20, and 30 cases, and the credible intervals of all parameter estimates overlapped (S1 Text).

Cities that did not meet the thresholds for cumulative reported cases were treated as missing in the analysis. Similar approaches have been employed in the study of seasonal and pandemic influenza [20,21,26]. Here, some unaffected cities were not invaded because they were not at risk (due to environmental factors). Of the cities that were at risk, some were invaded but others appeared to have escaped invasion by chance or other unexamined factors. Among cities that escaped invasion, there is also some probability that they had in fact been invaded but never reported cases. Alternative study designs would be more appropriate for determining why some cities appeared to escape invasion. For example, a mechanistic model of disease transmission accounting for environmental conditions such as temperature and rainfall could be used to ascertain why some cities were invaded and others were not [11]. Also, community-based studies conducted shortly after the epidemic could have assessed whether a city was invaded but did not report cases. Community-based studies were conducted in Colombia following both epidemics, but only in cities that reported many cases [29–31].

Ninety-nine percent of chikungunya fever cases and 95% of ZIKV disease cases were clinically confirmed, rather than laboratory confirmed. This could have led to misclassification, as dengue virus, CHIKV, and ZIKV were circulating simultaneously. All three arboviruses cause

similar symptoms which makes differential diagnoses based on clinical grounds alone a challenge in endemic settings [32]. Also, asymptomatic infection, mild illnesses, and limited access to healthcare likely resulted in underreporting. Problems with reporting and misdiagnoses may have affected the fit of the probability distribution of invasion week for cities invaded near the end of the CHIKV epidemic (Fig 4). Some of these late-invaded cities might have been invaded earlier but not reported cases to the surveillance system in a timely manner. Another possible explanation is that cases reported at the end of the CHIKV epidemic were actually misdiagnosed ZIKV disease cases. Oliviera et al. studied the interrelationships between cases of dengue fever, chikungunya fever, and ZIKV disease in Brazil from 2015–2017. Confirmed cases included all suspected cases reported to the national surveillance system, while discarded cases were defined as suspected cases that met at least one of the following conditions: (i) negative laboratory diagnosis by IgM serology, (ii) laboratory confirmation of another disease, and (iii) clinical and epidemiological compatibility with another disease. Using an autoregressive model, they found that the time series of confirmed and discarded cases of dengue fever significantly affected the time series of confirmed and discarded cases of ZIKV disease and the other way around. Although confirmed and discarded cases of chikungunya fever were found to affect the reporting of dengue fever, there was no evidence that the reporting of ZIKV disease or dengue fever affected reporting of chikungunya fever [33].

Historical dengue transmission in Colombia could have played a role in the spread of CHIKV and ZIKV. Although it is unlikely that high levels of dengue would have affected susceptibility to CHIKV, an unrelated alphavirus, there is some evidence of cross-reactivity among flaviviruses, such as ZIKV and dengue virus. If pre-existing immunity for dengue virus increased the risk of symptomatic ZIKV infection, we would expect faster recognition of ZIKV in cities that are hyperendemic for dengue. A cohort study in Managua, Nicaragua found evidence that prior dengue infection was protective for symptomatic ZIKV infection among children (incidence rate ratio 0.62, 95% CI: 0.44–0.86) adjusting for age, sex, and recent infection with dengue virus [34]. However, a cohort study in Salvador, Brazil found that individuals with high antibody titers to dengue virus had less risk of ZIKV infection and symptoms [35]. We found that high historical levels of dengue in a city decreased the time to invasion for both CHIKV and ZIKV, suggesting that other factors such as environmental suitability of *Aedes* mosquitoes are more important to city invasion than potential impacts of cross-reactive immunity among flaviviruses.

A further limitation is that the model only incorporates one distance metric at a time. In reality, the spread of ZIKV and CHIKV was likely driven by a combination of air travel, land-based travel, and vector movement. The model also does not consider changes over time in reporting, human behavior, or transmission. These aspects could have changed during the epidemics, especially when the Public Health Emergency of International Concern was declared by the World Health Organization in February 2016 [6].

Another assumption of this model is that CHIKV and ZIKV were each introduced into Colombia only once. The results of two recent genomic studies suggest that this assumption is valid which is why we did not find it necessary to account for background importation rates of CHIKV and ZIKV. Black et al. found evidence of two separate introductions of ZIKV into Colombia; however, the majority of cases were associated with a single introduction [28]. Similarly, Villero-Wolf et al. found evidence of only three introductions of CHIKV in Colombia, suggesting that most cases resulted from transmission within the country, rather than repeated travel-related importations [36].

The gravity model formulation used in this study works well retrospectively; however, more work is needed to understand why some cities appear to escape invasion. Until this issue is resolved, these methods have limited use for real-time forecasting of epidemics.

Future directions for this work include the use of this approach to understand the invasion dynamics of other epidemics. Further research should also focus on quantifying the relative contribution of human versus vector movement on spatial transmission. This would have broad implications for surveillance and control for other mosquito-borne epidemics such as dengue, Mayaro, and yellow fever.

## Methods

### Ethics statement

The technical and ethical endorsement of the study was provided by the Comité de Ética y de Metodologías de Investigación of Colombia's Instituto Nacional de Salud (project number 35–2017). Consent was not obtained from study participants as the data were analyzed anonymously.

### Data

We analyzed anonymized line list data on 105,152 ZIKV disease and 411,789 chikungunya fever suspected and laboratory-confirmed cases reported to Sivigila, Colombia's national public health surveillance system, between 2014 and 2017. See S1 Text for a full description of epidemiological and demographic data as well as data on elevation, weather, socioeconomic status, and a proxy of human mobility.

### Definition of invasion week

Out of 1,122 cities in Colombia, only cities with at least 20 cases of chikungunya fever were considered to have been "invaded" by CHIKV. Similarly, only cities with at least 30 cases of ZIKV disease were considered to have been "invaded" by ZIKV. Cities with case counts below these thresholds were not considered in the primary analysis. Invasion was defined as the week before cases were first reported in each city. We assumed a latent period of one week after which the city is considered infectious and can spread the infection to other cities. S1 Text includes details of and model fits to an alternative definition of invasion week.

### Potential sources of the epidemics

The cities in Colombia where the CHIKV and ZIKV epidemics most likely originated were identified. The method is based on the idea that epidemics spread radially from the origin, meaning that the relationship between invasion week and distance from the source is linear [21]. Assuming a single introduction of each virus into Colombia, the first 10% of invaded cities were considered as potential origins for the epidemics. Pearson's correlation coefficient for the relationship between the city's invasion week and its geographic distance to the origin was calculated for each potential origin. The most likely source was identified as the city with the highest such correlation coefficient (Fig 2). Reporting at the beginning of the epidemics would have been unreliable because the diseases were new to the country, there were a high proportion of asymptomatic infections, and the presentation of symptoms is fairly unspecific. Therefore, the line list data alone are unlikely to be very useful for determining the geographic origin of the epidemics.

### Long-distance transmission events

The number and location of long-distance transmission events of CHIKV and ZIKV were identified using the invasion week in each city. The method detects outliers in the distribution of pairwise distances between newly invaded cities and the set of infectious cities at the previous time step [21]. $C$ is defined as the set of cities in the network. At time $t_j$, $C$ is divided into

the set of infectious cities $I_{t_j}$ and the set of susceptible cities $S_{t_j}$:

$$I_{t_j} = \{k : t_k < t_j\}$$

$$S_{t_j} = \{k : t_k \geq t_j\}$$

where $t_j$ is the timing of invasion of city $j$ and $t_k$ is the timing of infectiousness (invasion week plus one week) in each of k cities. For city $j$, the minimum distance between city $j$ and infectious cities in $I_{t_j}$ was calculated as $d_j$, the most likely route of invasion when the spatial dynamics are dominated by distance.

$$d_j = \min_{k} d_{jk}$$

for $k \in I_{t_j}$. We also calculated $D_j$, the minimum distance between city $j$ and any other city in the network:

$$D_j = \min_{i} d_{ji}$$

for $i \in C$. If the process were entirely spatial, cities would usually be invaded by nearby cities. Thus, the distance to the nearest city is approximated by $d_j - D_j \approx 0$. For each city, $d_j - D_j$ was calculated and those included in the 99[th] percentile of the distribution of $d_j - D_j$ were considered long-distance transmission events (S1 Text).

## Spatial interaction models

We considered four main types of spatial interaction models, (i) the gravity model, (ii) the competing destinations model, (iii) Stouffer's rank model, and (iv) the radiation model. For each model type, we considered both geographic distance and travel time between cities.

Model parameters were initially estimated independently for each virus. From the four best-fitting models with a common structure, we then ran joint models assuming the same parameters across CHIKV and ZIKV. We also considered joint models in which some parameters were allowed to vary across viruses. From the first approach, we obtained the best-fitting model for each virus. From the second approach, we obtained the best-fitting joint model highlighting the commonalities in the spatiotemporal dynamics across CHIKV and ZIKV.

**Gravity models.**   We fitted gravity models to analyze transmission of CHIKV and ZIKV between cities that reported the minimum number of cases. These models have been used to study transmission of infectious diseases, such as measles [37], influenza [20,21,23,26], vector-borne diseases [38], and cholera [39,40], among others.

For each ZIKV and CHIKV separately, $N$ cities have an invasion week, $t_i$, which was defined as one week before cases were first reported in each city. Cities also have population size, $P_i$, which is assumed to be constant over time and weekly case counts weighted by the generation time distribution, $c_{i,t}$ (S1 Text). The generation time is the average time between the time of infection in a primary case and the time of infection in a secondary case infected by the primary case [41]. The geographic distance in km (or travel time in minutes) between invaded city $i$ and susceptible city $j$ is $d_{ij}$. For geographic distance, we used the geodesic distance on an ellipsoid, which is the shortest path between two points accounting for the curvature of the Earth. Further details are given in S1 Text.

At each time point, a city can be either "susceptible," "latently invaded," or "infectious." Once cities are invaded, they are latently invaded for one week and then infectious. After

external seeding into Colombia occurs, we assumed no additional cases are imported from abroad. If a city is invaded in week $t_i$, only Colombian cities that were infectious in the previous week could have spread the disease to that city. We assumed transmission parameters remain constant over time.

As in Eggo et al.[20], the force of infection, λ, represents the hazard of infection from an invaded city to a susceptible city. At time $t$, the force of infection from city $i$ to city $j$ can be defined as:

$$\lambda_{i \to j,t} = \beta c_{i,t}^{\phi} P_j^{\mu} \frac{P_i^{\upsilon}/d_{ij}^{\gamma}}{\left( \Sigma_{k,k \neq j} \frac{P_k^{\upsilon}}{d_{kj}^{\gamma}} \right)^{\varepsilon}}$$

Exponents ν and μ are for population sizes of city $i$ and $j$, respectively. The distance between cities is $d_{ij}$ and γ is the power parameter. β describes transmission intensity. φ captures the relationship between infectivity of a city and its weekly case count weighted by the generation time distribution. A value of φ = 1 indicates that the infectiousness of a city at time $t_i$ is proportional to the number of cases reported in that city weighted by the generation time distribution at time $t_i$. When φ = 0, infectiousness does not depend on the number of reported cases in the source city. Values of φ between 0 and 1 lead to infectiousness profiles that vary according to weekly case counts. Parameter ε characterizes the density dependence of the connection between a susceptible city and all invaded cities. When ε = 0, the formulation above reduces to a simple density-dependent model, and when ε = 1, the formulation above reduces to a density-independent model. When ε is estimated, the model is equivalent to Fotheringham's competing destinations model [19]. The total force of infection on city $j$ at time $t$ is defined by:

$$\lambda_{j,t} = \sum_{i \neq j}^{i} \lambda_{i \to j,t} I_{ij,t}$$

where 
$$I_{ij,t} = \begin{cases} 1, \text{if } i = \text{Infectious and } j = \text{Susceptible} \\ 0, \text{otherwise} \end{cases}$$

The probability that a susceptible city $j$ is invaded at time $t_j$ is

$$P(t_j) = exp\left( -\sum_{\tau=0}^{t_j-1} \lambda_{j,\tau} \right)\left( 1 - exp\left( -\lambda_{j,t_j} \right) \right)$$

The first part of the equation is the probability that a city escaped invasion from $t = 0$ until just before $t_j$. The second part is the probability that the city was invaded at $t_j$ given that it was susceptible until that week. The conditional log likelihood is summed over all susceptible cities:

$$l = \sum_j \ln\left( P(t_j) \right)$$

We first investigated null models that only included β. Parameters were then added to test for a spatial effect in transmission, the role of population size of invaded and susceptible cities, and infectivity. Except for β, which is always estimated by Markov chain Monte Carlo (MCMC), parameters can be fixed at 0, at 1 (not γ), or estimated by MCMC.

**Stouffer's rank model.** According to Stouffer's rank model, also known as the law of intervening opportunities, the number of people traveling a particular distance is proportional

to the number of opportunities at that distance and inversely proportional to the number of opportunities along the way [42]. Following [22], we used population size as a proxy for "opportunities." Using our notation, the force of infection from city $i$ to city $j$ at time $t$ is

$$\lambda_{i \to j,t} = \beta c_{i,t}^{\phi} P_j^{\mu} \left( \frac{P_i}{\sum_{k \in \Omega(j,i)} P_k} \right)^{\upsilon}$$

where $k \in \Omega(i,j)$ is the group of cities that are closer to susceptible city $j$ than invaded city $i$: $\Omega(j,i) = \{k: 0 < d(j,k) \leq d(j,i)\}$. We also considered a variant of this model in which city $j$ is included among the intervening opportunities. In "Stouffer's rank variant model," $\Omega(j,i) = \{k: 0 \leq d(j,k) \leq d(j,i)\}$ which allows within-city opportunities to decrease spatial coupling.

**Radiation model.** We also considered the radiation model which was proposed by Simini et al. [43]. This model also accounts for higher-order interactions among population centers and is related to Stouffer's rank model. Our version of the radiation model is

$$\lambda_{i \to j,t} = \beta c_{i,t}^{\phi} P_i \frac{P_i P_j}{(P_i + \sum_{k \in \Omega(i,j)} P_k)(P_j + P_i + \sum_{k \in \Omega(i,j)} P_k)}$$

where again we considered two variants, one in which city $j$ is excluded ("radiation") from the set $\Omega(j,i)$ and one in which it is included ("radiation variant"). The radiation models do not have a parameter for the spatial component.

## Model estimation and computing

Metropolis-Hastings MCMC sampling was used to investigate the posterior distributions of parameters [44,45]. Because the parameters cannot take negative values, we sampled from a log normal distribution and corrected the Metropolis accept-reject rule for asymmetric jumping. Parameters were updated one at a time. Uniform prior distributions were used for all parameters. Three chains were run for each model with different starting values, and chains were visually checked for convergence after 100,000 iterations with a burn-in of 0.2 times the length of the chains (iterations times number of parameters). We also used the coda package (version 0.19–4) in R to calculate the Gelman-Rubin statistic for each best-fitting model. This statistic assesses model convergence by comparing the variance between- versus within-MCMC chains. Lack of convergence is indicated by values above one [46,47] (S1 Text). Median parameter estimates and 95% credible intervals were calculated from the posterior distributions after excluding the burn-in.

DIC was used to compare models. Lower values of DIC are preferred, and a difference of about 5 is considered meaningful [48]. DIC was calculated using the medians of the posterior distributions of the parameters due to non-normality of the likelihood.

All analyses were performed in R version 3.5.1. Data and code for reproducing the best-fitting gravity model results and the figures in the main text as well as the weekly time series of reported chikungunya fever and ZIKV disease cases at the city level are available on GitHub (http://github.com/kcharniga/zika_chik_invasion).

## Validation of gravity model fit and sensitivity analyses

The probability distribution of the invasion week was calculated for each city based on the observed start of invasion in other cities up to that time. For each virus, this calculation was performed using the median parameter estimates from the posterior distribution. The probability distributions were then compared with the observed invasion weeks.

We also simulated the CHIKV and ZIKV epidemics in Colombia using 1,000 parameter sets sampled from the posterior distribution. For each set, the city (or cities) invaded in the first time step served as the origin of the epidemic. A random deviate was chosen from a uniform distribution between 0 and 1 for each city in each week. If the probability of $t_j$ was higher than the random deviate, the city became invaded. Once invaded, the observed weekly case counts weighted by the generation time distribution were used to model that city's infectiousness over time. Epidemic simulations were also used to test the sensitivity of the case count thresholds used to determine the number of invaded cities for each virus. Epidemic simulations were also performed for the best-fitting Stouffer's rank models (S1 Text).

## Validation of parameter fitting procedure

Using the framework above, we validated the fitting procedure for the model parameters by simulating one dataset for each virus with the median parameter estimates obtained from the best-fitting models. The analysis was re-run on each simulated dataset to check that the fitted parameter estimates could be recovered.

## Risk factors of invasion

Logistic regression models were used to determine risk factors for invasion by CHIKV and ZIKV. The outcome was defined as a city reporting at least 20 cases of chikungunya fever and at least 30 cases of ZIKV disease for each respective model. Predictors included population size, elevation, dengue risk, temperature, rainfall, and mean travel time as well as the percentage of households in each city with overcrowding, inadequate exterior walls, and inadequate flooring. Dengue risk was categorized into four levels as follows: cities located at or below 1800 m of elevation that reported any cases of dengue fever between 2010–2016 were considered "at risk" of dengue. The natural logarithm of the cumulative number of cases over this period was taken and divided into tertiles (1–3 with 3 being the highest). All other cities were assigned values of 0. Mean temperature for each city was obtained by taking the mean of the weekly population-weighted weekly time series of mean temperature over the study period, defined as the time during which cases were being reported in the country (110 weeks for CHIKV and 97 weeks for ZIKV). Similarly, mean rainfall was calculated for each city as the mean of the population-weighted weekly time series of cumulative precipitation for each respective study period. As a sensitivity analysis, we also considered the mean of the weather covariates from the week cases were first reported until the peak of each epidemic (34 weeks for CHIKV and 26 weeks for ZIKV). Mean travel time for each city was defined as the average time to travel from that city to all other cities, excluding the islands of San Andrés and Providencia. Further details about the data can be found in S1 Text.

The predictors were first explored in a univariate analysis (S1 Text). Significance of difference between invaded and uninvaded cities was tested by chi-square tests for categorical variables and Wilcoxon rank-sum test for continuous variables, none of which were normally distributed. P values < 0.05 were considered statistically significant. A forward stepwise approach was then used to build each logistic regression model: predictors were added to the model one at a time and only kept if they were significant at the 0.05 level. The units of rainfall and elevation were changed to 10 mm and 100 m, respectively, to improve the interpretation of the odds ratios which were computed by exponentiating model coefficients. Models testing the effect of mean travel time were fitted to 1,120 cities for which data were non-missing. The Hosmer and Lemeshow goodness-of-fit test was applied to the best-fitting models using 10 groups.

Linear regression was also performed to assess the relationship between dengue risk and time of invasion for CHIKV and ZIKV.

## Supporting information

**S1 Text. Fig A. Weekly reported cases of chikungunya fever (CF), dengue fever (DF), and Zika virus disease (ZVD) in Colombia, January 2010 –June 2017. Fig B. Epidemiological curves of chikungunya fever (CF) and Zika virus disease (ZVD) in Colombia by department, 2014–2017.** Departments are ordered from North to South down the columns. Y axes are different for each plot. **Fig C. City elevation.** Comparison of elevation (in meters) between cities that were invaded versus cities that escaped invasion for (a) CHIKV, (b) ZIKV, and (c) CHIKV, ZIKV, or DENV. **Fig D. Example of algorithm used to estimate invasion week using the generation time method.** (a) The time series for Caucasia in the department of Antioquia during the 2015–2017 ZIKV epidemic. In this figure, week 1 corresponds to the week ending on August 15, 2015, and week 51 corresponds to the week ending on July 30, 2016. The algorithm identifies the point of maximum incidence in the time series and counts backward one week at a time until there are no reported cases. If there are no cases in this week or the prior two or three weeks depending on the infection's generation time, then this is the invasion week. If not, the algorithm continues to go back in time until the condition is met. The part of the line in red is the period used to determine the onset of invasion, and the blue dashed line is the estimated invasion week. (b) The same time series as in (a) is shown until the point of maximum incidence (week ending on January 23, 2016). The estimated time of invasion is week 15 rather than week 21 because cases were reported in weeks 16–20. **Fig E. Comparison of estimated invasion weeks using two methods.** A method based on the first reported cases in each city (x-axis) and a method based on the generation time distribution of each infection (generation time method, y-axis) were compared for (a) CHIKV and (b) ZIKV. The black line is y = x. The two methods show good agreement (CHIKV: r = 0.60, ZIKV: r = 0.68). **Fig F. Comparison of estimated invasion weeks using two methods.** A method based on the generation time distribution (generation time method, x-axis) and a piecewise spline method (Charu method, as in [21], y-axis) were compared for (a) CHIKV and (b) ZIKV. 95% confidence intervals are shown for the Charu method only. For some cities, only the point estimate for $t_j$ fell within the 95% confidence interval; this is shown by a lack of vertical bar. The two methods show very good agreement (CHIKV: r = 0.90, ZIKV: r = 0.70). **Fig G. Distribution of invasion week by dengue risk level for (a) CHIKV and (b) ZIKV.** The black lines are the fitted linear regression models. **Fig H. MCMC chains for the best-fitting CHIKV gravity model from three different starting points. Fig I. MCMC chains for the best-fitting ZIKV gravity model from three different starting points. Fig J. Epidemic simulations of the best-fitting gravity models showing the sensitivity of the thresholds used to determine invasion. Fig K. Comparison of the distance kernel obtained when running the CHIKV gravity model from week 12 versus the entire dataset.** The distance power estimates were similar when parameter estimation started from week 12 (1.77 [95% CrI: 1.54–1.99]) compared to week 1 (1.68 [95% CrI: 1.44–1.90]). **Fig L. Probability distribution of estimated invasion week (generation time method).** Colored lines represent the probability distribution of invasion week for (a) CHIKV and (b) ZIKV. The calculations were performed using the median parameter estimates from the posterior distributions of the models using estimated invasion week rather than first reported cases. The black lines show the estimated invasion week in each city using a method based on each infection's generation time. Values of 0.01 represent probabilities of 0.01 or less. **Fig M. Epidemic invasion simulations (generation time method).** Simulated invasion for (a) CHIKV and (b) ZIKV from the models using estimated

invasion week by generation time method rather than week of first reported cases. Simulated epidemics are shown in light gray. The dark gray lines are the average across the 1,000 simulations. The red lines are the observed incidence curves. **Fig N. Epidemic invasion simulations (all cities).** Simulated invasion for (a) CHIKV and (b) ZIKV from the models using week of first reported cases and all 1,122 cities in Colombia. Simulated epidemics are shown in light gray. The dark gray lines are the average across the 1,000 simulations. The red lines are the observed incidence curves. **Fig O. Epidemic invasion simulations (best-fitting Stouffer's rank models).** Results correspond to the models presented Tables S-T. Simulated invasion for (a) CHIKV and (b) ZIKV from the models using week of first reported cases. Simulated epidemics are shown in light gray. The dark gray lines are the average across the 1,000 simulations. The red lines are the observed incidence curves. **Fig P. Long-distance transmission events.** The distribution of d-D for (a) CHIKV and (b) ZIKV in this study. The dashed blue lines are plotted at the 97.5[th] percentile (corresponding to 212.00 km and 255.33 km for CHIKV and ZIKV, respectively) and the dashed red lines are plotted at the 99[th] percentile (corresponding to 344.40 km and 321.21 km for CHIKV and ZIKV, respectively). Long-distance transmission events were defined as invasions that occurred in cities included in the 99[th] percentile of this distribution. **Fig Q. Probability distribution of first reported cases by department for CHIKV.** Black circles are cities that fall within the 95% interval of their expected distribution, and red circles fall outside this interval. The gray circle in the department of Córdoba represents the city that was invaded in week 0. **Fig R. Probability distribution of first reported cases by department for ZIKV.** Black circles are cities that fall within the 95% interval of their expected distribution, and red circles fall outside this interval. The gray circles in the departments of San Andrés, Valle del Cauca, and Norte de Santander represent cities that were invaded in week 0. **Fig S. Parameter estimates for CHIKV model fitted using estimated invasion week by generation time method.** The dashed red line shows the median of the posterior distribution of each parameter after removing the burn-in period. The blue line shows the median of the posterior distribution from the model fitted using the method based on first reported cases as in the main text. Only parameters that are estimated in both models are shown. **Fig T. Parameter estimates for ZIKV model fitted using estimated invasion week by generation time method.** The dashed red line shows the median of the posterior distribution of each parameter after removing the burn-in period. The blue line shows the median of the posterior distribution from the model fitted using the method based on first reported cases as in the main text. Only parameters that are estimated in both models are shown. **Table A. Gelman-Rubin statistic for each of the best-fitting gravity models (after removing the burn-in). Table B. Acceptance percentages for parameters of the best-fitting CHIKV and ZIKV gravity models. Table C. Summary statistics of the d-D distributions.** This table shows that ZIKV and CHIKV exhibited similar patterns of transmission. The first six columns have units in km. The seventh column is the total sample size, and the last two columns contain the number of long-distance transmission events for two distance thresholds. **Table D. Recipient and potential source cities of long-distance transmission events of CHIKV. Table E. Recipient and potential source cities of long-distance transmission events of ZIKV. Table F. Estimates of the mean and standard deviation of the generation time distribution.** Estimates were used to calculate city infectivity. All values have units in days. **Table G. Univariate analysis of risk factors of CHIKV invasion. Table H. Univariate analysis of risk factors of ZIKV invasion. Table I. Best-fitting logistic regression model of CHIKV invasion.** Models were fitted to 1,120 cities because mean travel time between cities was not available for two island cities. **Table J. Best-fitting logistic regression model of ZIKV invasion.** Models were fitted to all 1,122 cities in Colombia. **Table K. Comparison of parameter estimates from gravity models fitted to different numbers of cities using thresholds of

**10, 20, and 30 cumulative reported cases.** In each case, the model is the infectivity model with μ set to 0. Columns in bold correspond to results presented in the main text. **Table L. Comparison of parameter estimates from observed data versus simulated data. Table M. Parameter estimates for six gravity models of CHIKV for 337 cities using geographic distance.** Posterior median and 95% credible interval presented for each parameter. Bold indicates the best-fitting model. Travel time data were only available for 337 out of 338 cities. To compare across distance metrics, 337 cities were also used for geographic distance models. **Table N. Parameter estimates for six gravity models of CHIKV for 337 cities using travel time between cities.** Posterior median and 95% credible interval presented for each parameter. Bold indicates the best-fitting model. Travel time data were only available for 337 out of 338 cities. **Table O. Parameter estimates for six gravity models of ZIKV for 287 cities using geographic distance.** Posterior median and 95% credible interval presented for each parameter. Bold indicates the best-fitting model. Travel time data were only available for 287 out of 288 cities. To compare across distance metrics, 287 cities were also used for geographic distance models. **Table P. Parameter estimates for six gravity models of ZIKV for 287 cities using travel time between cities.** Posterior median and 95% credible interval presented for each parameter. Bold indicates the best-fitting model. Travel time data were only available for 287 out of 288 cities. **Table Q. Comparison of alternative models of CHIKV and ZIKV spread in Colombia.** Posterior median and 95% credible interval presented for each parameter. Models are ordered by sum of DIC and were fitted separately to 337 cities for CHIKV and 287 cities for ZIKV. **Table R. Comparison of individual versus joint models of CHIKV and ZIKV spread in Colombia (third and fourth best-fitting model variants).** Posterior median and 95% credible interval presented for each parameter. Models were fitted to 337 cities for CHIKV and 287 cities for ZIKV. For the joint models that estimate different parameters across arboviruses, parameters with subscript a refer to CHIKV, while parameters with subscript b refer to ZIKV. **Table S. Parameter estimates for seven models of CHIKV in Colombia for 338 cities.** The first six models are variations of the gravity model. Posterior median and 95% credible interval presented for each parameter. Bold indicates the model used in the validations. **Table T. Parameter estimates for seven models of ZIKV in Colombia for 288 cities.** The first six models are variations of the gravity model. Posterior median and 95% credible interval presented for each parameter. Bold indicates the model used in the validations. (DOCX)

**S1 Movie. Monthly chikungunya fever incidence per 100,000 population by department on a hexagonal grid.**
(MOV)

**S2 Movie. Monthly ZIKV disease incidence per 100,000 population by department on a hexagonal grid.** For both movies, the following abbreviations were used for department names: AMA = Amazonas, ANQ = Antioquia, ARA = Arauca, ANT = Atlántico, BOL = Bolívar, BOY = Boyacá, CAL = Caldas, CAQ = Caquetá, CAS = Casanare, CAU = Cauca, CES = Cesar, CHO = Chocó, COR = Córdoba, CUN = Cundinamarca, GUA = Guainía, GUV = Guaviare, HUI = Huila, LAG = La Guajira, MAG = Magdalena, MET = Meta, NAR = Nariño, NDS = Norte de Santander, PUT = Putumayo, QUI = Quindío, RIS = Risaralda, SAA = San Andrés and Providencia, SAN = Santander, SUC = Sucre, TOL = Tolima, VDC = Valle del Cauca, VAU = Vaupés, VIC = Vichada. Note that the island department of San Andrés and Providencia is attached to the mainland on the left side of the map.
(MOV)

## Acknowledgments

The authors would like to thank all of the medical and public health professionals involved in the reporting of chikungunya fever and ZIKV disease cases to Sivigila, Colombia's national public health surveillance system.

## Author Contributions

**Conceptualization:** Pierre Nouvellet, Christl A. Donnelly.

**Data curation:** Zulma M. Cucunubá, Marcela Mercado, Franklyn Prieto, Martha Ospina.

**Formal analysis:** Kelly Charniga.

**Investigation:** Kelly Charniga, Zulma M. Cucunubá, Pierre Nouvellet.

**Methodology:** Pierre Nouvellet, Christl A. Donnelly.

**Software:** Kelly Charniga, Zulma M. Cucunubá, Pierre Nouvellet.

**Supervision:** Pierre Nouvellet, Christl A. Donnelly.

**Validation:** Kelly Charniga, Pierre Nouvellet.

**Visualization:** Kelly Charniga, Zulma M. Cucunubá.

**Writing – original draft:** Kelly Charniga.

**Writing – review & editing:** Kelly Charniga, Zulma M. Cucunubá, Marcela Mercado, Franklyn Prieto, Martha Ospina, Pierre Nouvellet, Christl A. Donnelly.

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
