## [Decision Letter · Decision Letter 0]

9 Dec 2020

Dear Ms. Charniga,

Thank you very much for submitting your manuscript "Spatial and temporal invasion dynamics of the 2014-2017 Zika and chikungunya epidemics in Colombia" for consideration at PLOS Computational Biology.

As with all papers reviewed by the journal, your manuscript was reviewed by members of the editorial board and by several independent reviewers. In light of the reviews (below this email), we would like to invite the resubmission of a significantly-revised version that takes into account the reviewers' comments.

In particular, Reviewer 1 raises important issues about the robustness results given that only one spatiotemporal model was considered and no comprehensive validation of the inference framework performed. We would expect these to be addressed in a revised version. Moreover, please provide some documentation with the code provided as it is not currently clear what steps need to be taken to reproduce the results presented in the manuscript. Please also refer to the code/data supplement in the main manuscript, as one of the reviewers was not aware that data/code had been provided.

We cannot make any decision about publication until we have seen the revised manuscript and your response to the reviewers' comments. Your revised manuscript is also likely to be sent to reviewers for further evaluation.

Sincerely,

Sebastian Funk

Guest Editor

PLOS Computational Biology

Thomas Leitner

Deputy Editor

PLOS Computational Biology

Reviewer's Responses to Questions

**Comments to the Authors:**

Reviewer #1: Review is uploaded as an attachment.

Reviewer #2: "What are the main claims of the paper and how significant are they for the discipline?"

The aim of the work is to analyse the invasion and spread of Zika and chikungunya viruses in Colombian cities. To approach the problem the study relies on weekly reported cases of the two arboviruses for each of the cities considered in the study and the application of gravity models. The analysis estimates the geographic origin of both epidemics in the country. Moreover, the results show that the transmission spread is more likely to be driven by short distance transmission and potentialized by large populations, with Zika virus dissemination being faster than chikungunya. The results have a significant impact on the local and national level of surveillance systems and can be aggregated as a guide for future measures prevention and control of the diseases.

"Are these claims novel? If not, which published articles weaken the claims of originality of this one?"

The methodology and hypothesis in the work have been explored for other diseases and in other scenarios, which is well presented in the references provided by the authors. Nevertheless, these results go beyond previous reports, showing new perspectives for the emergence and spread of transmission of ZIKV and CHIKV in Colombia.

"Are the claims properly placed in the context of the previous literature?"

Yes.

"Have the authors treated the literature fairly?"

Discussion would require citation. See additional comments below.

"Do the data and analyses fully support the claims? If not, what other evidence is required?"

Yes.

"Would additional work improve the paper? How much better would the paper be if this work were performed and how difficult would it be to do this work?"

No

"Are original data deposited in appropriate repositories and accession/version numbers provided for genes, proteins, mutants, diseases, etc.?"

No.

"Does the study conform to any relevant guidelines such as CONSORT, MIAME, QUORUM, STROBE, and the Fort Lauderdale agreement?"

Yes.

"Are details of the methodology sufficient to allow the experiments to be reproduced?

Is any software created by the authors freely available?"

The methodology is well explained, however there is no data and code availability to allow reproducibility of the results.

"Is the manuscript well organized and written clearly enough to be accessible to non-specialists?"

Yes

"Does the paper use standardized scientific nomenclature and abbreviations? If not, are these explained at the first usage?"

Yes.

It follows down some additional comments.

Comment 1: The authors must explain the use of the data of dengue fever cases mentioned in the supplementary material.

Comment 2: Does the invasion of ZIKV and CHIKV have any influence caused by the historical dengue virus transmission in the country? Do the hyperendemic cities for DENV have any influence on the delay of invasion by these arboviruses? Is there a possibility to discuss that?

Comment 3: It would be interesting to add a plot (perhaps on supplementary material) of weekly series of cases of ZIKV, CHIKV (and possibly DENV), for the period of study for the country level (or other appropriate regional divisions as desired by the authors).

Comment 4: Is there the possibility to combine Figure 3 in order to better visualize spatial and temporal occurrences of cases (incidence) of the diseases? Is there any situation of a peak of both diseases occurring at the same time and space?

Comment 5: Reference is needed to foment the discussion presented from line 299 to 307. An example can be found in a recent work that analysed misclassification due to the simultaneous co-circulation of DENV, CHIKV, and ZIKV, in Brazil link https://doi.org/10.1371/journal.pone.0228347 .

**Have all data underlying the figures and results presented in the manuscript been provided?**

Reviewer #1: None

Reviewer #2: **No: **The methodology is well explained, however there is no data and code availability to allow reproducibility of the results.

PLOS authors have the option to publish the peer review history of their article (what does this mean?). If published, this will include your full peer review and any attached files.

Reviewer #1: No

Reviewer #2: No
---

## [Decision Letter · Decision Letter 1]

27 May 2021

Dear Ms. Charniga,

We are pleased to inform you that your manuscript 'Spatial and temporal invasion dynamics of the 2014-2017 Zika and chikungunya epidemics in Colombia' has been provisionally accepted for publication in PLOS Computational Biology.

Best regards,

Sebastian Funk

Guest Editor

PLOS Computational Biology

Thomas Leitner

Deputy Editor

PLOS Computational Biology

Reviewer's Responses to Questions

**Comments to the Authors:**

Reviewer #1: The authors have done an impressive amount of work to revise their manuscript. They have satisfied the concerns that I had, particularly around the choice of mobility model and the need for simulation studies to validate their method.

My only reservation of the method is its inability to explain the lack of invasion of Zika in some administrative units. However, I believe that the authors have provided sufficient discussion of this limitation. It may be that a more complicated model structure that allowed transmission rates to vary across locations would be needed to fully explain the data. However, I think that by conditioning on the locations that had an invasion during the study period, the analysis is appropriate.

Reviewer #2: The answers are satisfactory.

**Have the authors made all data and (if applicable) computational code underlying the findings in their manuscript fully available?**

Reviewer #1: Yes

Reviewer #2: Yes

PLOS authors have the option to publish the peer review history of their article (what does this mean?). If published, this will include your full peer review and any attached files.

Reviewer #1: No

Reviewer #2: No

---

## [Editor Report · Acceptance letter]

28 Jun 2021

PCOMPBIOL-D-20-01607R1 

Spatial and temporal invasion dynamics of the 2014-2017 Zika and chikungunya epidemics in Colombia

Dear Dr Charniga,

I am pleased to inform you that your manuscript has been formally accepted for publication in PLOS Computational Biology. Your manuscript is now with our production department and you will be notified of the publication date in due course.

With kind regards,

Zsofi Zombor
